# Macroinvertebrate habitat requirements in rivers: overestimation of environmental flow calculations in incised rivers

Renata Kędzior[1], Małgorzata Kłonowska-Olejnik[2], Elżbieta Dumnicka[3], Agnieszka Woś[4], Maciej Wyrębek[4], Leszek Książek[4], Jerzy Grela[5], Paweł Madej[5], Tomasz Skalski[6]

[1]Department of Ecology, Climatology and Air Protection, Faculty of Environmental Engineering and Land Surveying, Agricultural University of Krakow, 30059, Krakow, Poland
[2]Centre of Research and Science Innovations, 20819, Lublin, Poland
[3]Institute of Nature Conservation, Polish Academy of Science, 31120, Krakow, Poland
[4]Department of Hydraulic Engineering and Geotechnics, Faculty of Environmental Engineering and Land Surveying, Agricultural University of Kraków Poland, 30059, Krakow, Poland
[5]MGGP joint-stock company, 33100, Tarnów, Poland
[6]Biotechnology Centre, Silesian University of Technology, 44100 Gliwice, Poland

*Correspondence to*: Tomasz Skalski (tomasz.skalski@polsl.pl)

**Abstract.** Flow variability determines the conditions of river ecosystem and river ecological functioning. The variability of ecological processes in river ecosystems gradually decreases due to river channelization and incision. Prediction of the environmental flow allowing to keep biological diversity and river health developed as a response to the degradation of aquatic ecosystems overexploited by human. The goal of the study was to test the influence of river incision on environmental flow estimation based on the biological monitoring working party macroinvertebrate index. The 240 macroinvertebrate assemblages of 12 waterbodies differing in the bed substrate, amplitude of discharge were surveyed in southern Poland. The variations in the distribution of 151 466 macroinvertebrates belonging to 92 families were analysed. The similarity of benthic macroinvertebrates reflects the typological division of the rivers into three classes: mountain Tatra streams, mountain flysch rivers, and upland carbonate and silicate rivers. As a response variable reflecting the macroinvertebrate distribution in the river, environmental parameters, BMWP_PL index was chosen. The river incision significantly increased the values of e-flow calculations in relation to redeposited channels. The area of optimal habitat for macroinvertebrates decreased with the bed incision intensity. In highly incised rivers, the environmental flow values are close to the mean annual flow, suggesting that a high volume of water is needed to obtain good macroinvertebrate conditions. As a consequence, the river downcutting processes and impoverishment of optimal habitats will proceed.

## 1 Introduction

Human water demand, including irrigation to increase crop productivity, dams, and reservoirs to control the timing of stream flow, and water withdrawal from rivers, has increased dramatically over the last 100 years (Vörösmarty et al., 2010; Veldkamp et al., 2017). Maintenance of a suitable water flow in an active river channel should not only secure human needs, but above all ensure the proper functioning of aquatic ecosystems (Anderson et al., 2006). This has become particularly important since river beds began to be perceived not only as channels filled with water, but as complex ecological systems, in which biological elements play a key role (Poff et al., 1997; Bunn and Arthington, 2002; White et al., 2016). The Water Framework Directive (WFD, European Community, 2000/60/EC) was introduced by European countries to protect and improve the state of aquatic ecosystems and formalize a water flow framework that would maintain this state (Chen and Olden, 2017).

Discharge intensity is one of the most important factors influencing communities of aquatic and water-dependent organisms (Tharme, 2003; Arthington et al., 2006; Higgisson et al., 2019). It is a parameter which shapes the morphology (Michalik and Książek, 2009) and hydraulic flow conditions (water depth, flow velocity) and it influences the diversity and quality of habitats for fauna and flora in the active channel and in the floodplain (Allan, 1995; Poff et al., 1997; Ward and Tockner, 2001; Skalski

et al., 2016; 2020). Furthermore, flow significantly influences abiotic elements, such as water temperature and oxygenation,
as well as nutrient cycles in the aquatic ecosystem (Monk et al., 2008; Laini et al., 2019). This applies in particular to rivers
subjected to strong human impact (e.g., channel regulation and incision, dams, or retention reservoirs, as well as a continuous
increase in water abstraction). Artificial restriction and control of a range of water flow values leads to substantial
impoverishment of biological diversity (Pander et al., 2019). Environmental Flow is an amount of water required to maintain
biological diversity in the river ecosystem (Arthington et al., 2006). This definition requires to quantify ecological response of
aquatic elements to flow alteration, which data are rather scare in the literature (Poff and Zimmerman, 2010). Therefore, it
appears crucial to estimate empirical ranges of environmental flows that ensure optimal habitat conditions for living organisms
(Bunn and Arthington, 2002; Acreman et al., 2014).
Environmental flow has been studied by many researchers, resulting in numerous methods for its determination. The simpler
ones include hydrological methods, which are based on historical hydrological data and mean annual discharge (Tennant,
1976; Jowett, 1997; Tharme, 2003; Rosenfeld, 2017). Analysis of such data makes possible to specify a percentage of the
mean annual flow as the critical value below which severe degradation of biotic elements occurs. Unfortunately, hydrological
methods do not take into account the morphology of the river bed, which is a key factor shaping the river habitat (Książek et
al., 2020). Therefore, a number of hydraulic methods based on simple hydraulic variables such as critical riffle analysis
andwetted area/wetted perimeter have been introduced (Gippel and Stewardson, 1998; Książek et al., 2019). Determination
discharge values (Q) for environmental flow involves defining the breaking point of the hydraulic variable discharge curves
as the e-flow (Gippel and Sterwardson, 1998; Vezza et al., 2012; Tare at al., 2017). Over time, hydraulic methods have
developed in the direction of habitat simulation methods. They have additionally focused on the habitat requirements of
selected groups of model organisms, most commonly water depth, flow velocity, and bed substrate (Jowett and Davey, 2007;
Li et al., 2009; Muñoz-Mas et al., 2016). Based on the analysis of these environmental factors, habitat-discharge curves were
drawn for organisms, and from these it was possible to read the optimal flows maintaining the normal ecological functions of
aquatic ecosystems. Another type of method, which emphasizes the importance of the natural flow regime for the entire
ecosystem, are holistic methods. They attempt to maintain the natural flow regime as well as flow variability. In this case,
environmental flow is defined in the category of deviation from the natural flow regime (Yarnell et al., 2015).
The methods presented above focus on the fish distribution and rarely on diversity and availability of habitats for freshwater
macroinvertebrates, which are the most important and sensitive indicators of the ecological state of the ecosystem (Jowett et
al., 2008; Birk et al., 2012). The diversity and taxonomic composition of aquatic organisms living in freshwater streams and
rivers are used as indicators in the evaluation of environmental flow (Pander et al., 2019). In many cases, macroinvertebrate
assemblages are considered (Hayes et al., 2014; Laini et al., 2019), as numerous studies confirm that they are relatively good
indicators of ecological water quality and integrity (Buss et al., 2015; Wyżga et al., 2016; Schneider and Petrin, 2017).
Freshwater macroinvertebrates also play an important role in the processing of nutrients and organic energy in running water
ecosystems, as well as in sustaining ecosystem integrity.
Another parameter, which is usually neglected in flow modelling, is associated with morphological channel modification and
incision (Wyżga et al., 2012; Skalski et al., 2016). Incision and channel simplification is a global problem overwhelming most
of the rivers in the mountain as well as in upland areas (Skarpich et al., 2020). During the last 100 years anthropogenic
processes related to river regulation (narrowing and straitening) disturbed the fluvial processes leading to enormous river
incision (Rinaldi et al., 2005; Wyżga, 2008). As a results rivers become a vertically closed systems losing the ability to store
alluvial material. Moreover incision up to the bedrock simplifies the microhabitat array of the river (Neachell, 2014) and lead
to elimination most of the habitats (Muñoz-Mas et al., 2016) as well as affect ecosystem functioning (biodiversity lost and
food web network simplification, Shields et al., 1998; Jeffres et al., 2008).
The goal of the study was to test the influence of river incision on environmental flow estimation based on the biological
monitoring working party macroinvertebrate index. Specific aims of the study were: (1) to establish the habitat preferences of
macroinvertebrates communities (240 local assemblages) in mountain and upland rivers using generalized additive models,
(2) to calculate the e-flow values combining the habitat requirements and hydraulic method of environmental flow calculation
in relation to river hydromorphological parameters (redeposition and incision), (3) to identify reality of providing e-flow values
for different hydromorphological modifications in relation to available amount of water (Low Low Flow, Mean Low Flow and
Mean Annual Flow) and  (4) to check and visualize the e-flow values in relation to available water volume on randomly chosen,
incised, and redeposited rivers based on CCED2D model. We expected that e-flow in incised rivers, allowing to obtain the
shelf zone level of the river should be much higher than Mean Low Flow. Such assumption could determine the consecutive
higher discharges and increase the bed degradation. Firstly, we should restore the sedimentation processes in incised rivers to
obtain a hydrodynamic balance and then manage the proper volume of water. As a consequence, optimal habitats for
invertebrates and fish will be enlarged.
**2. Materials and methods**
**2.1 Study sites**
The survey was conducted in 12 mountainous rivers assigned to three typological groups according to the Polish Water
National Authority and the Water Framework Directive (Jusik et al., 2014): Tatra mountain rivers (Biały Dunajec, Dunajec,
and Białka - Group 1), mountain flysch rivers (Raba, Brynica, Toszecki Potok, and Nysa Kłodzka - Group 2) and upland
carbonate and silicate rivers (Sołokija, Warta, Ropa, Biała, and Odra - Group 3) (Fig. 1), varying in bed modification (incision
intensity or redeposition).

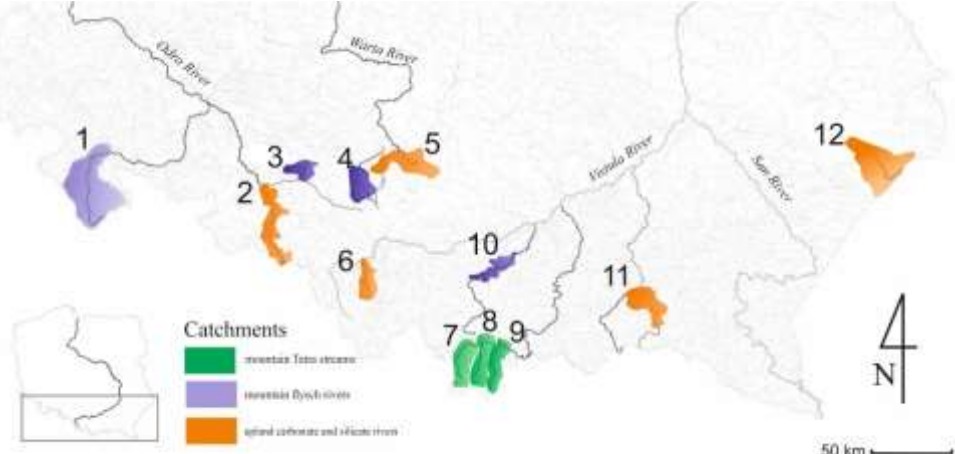


Figure 1. Map of the studied mountainous rivers in Carpatho-Sudetian region of Poland.

The first group comprises rivers located in an alpine granitoid region, characterized by calcareous and silicate bedrock. The
second group consists of rivers flowing through much lower mountain ranges (up to the timber zone), where the bedrock
contains sandstone rock formations. The third group represents rivers of upland landforms with various carbonate and silicate
sediments and rocks. The typology of river channel modification was obtained from field observation and channel
measurements (cross-sections, longitudinal profile and cover, high of the floodplain). Narrow channels with downcutting to
the floodplain and simplified channel morphology ware defined as incised.
All rivers are routinely monitored by the nearest monitoring station of the Environmental Agency (Environmental Agency
Data, 2018), and all twelve rivers have consistently been assigned a similar average chemical status in recent years. ANOVA
showed no variation between the river groups in incision bed modification (F=1.56, p=0.26) as well as in physicochemical
properties: dissolved oxygen, conductivity, hardness, $pH_{max}$, $NH_3$, $NO^{3-}$, $NO^{2-}$, total N, and $PO_4^{3-}$. Only water temperature and
pH min significantly depended on the river group. All habitat variables (flow, depth and substrate type) were significantly
dependent on river group (Table 1), meanwhile the incision was not influenced by the parameters variation.
Table 1 Mean values ± standard deviation of the physicochemical and habitat variables of the three river groups, with results
of one-way ANOVA.

| Environmental data | Group 1 | | Group 2 | | Group 3 | | F | p |
|---|---|---|---|---|---|---|---|---|
| | Mean | St. dev. | Mean | St. dev. | Mean | St. dev. | | |
| **Physicochemical** | | | | | | | | |
| Water temperature [°C] | 7.27 | 1.55 | 11.40 | 2.43 | 12.17 | 0.89 | 6.76 | **0.016** |
| Dissolved oxygen [mgL$^{-1}$] | 10.73 | 0.45 | 9.33 | 1.34 | 9.15 | 0.79 | 2.39 | 0.150 |
| Conductivity [$\mu$S cm$^{-1}$] | 202.67 | 91.58 | 1095.60 | 1594.59 | 356.5 | 93.26 | 0.85 | 0.458 |
| Water hardness [mg CaCO$_3$/l] | 113.00 | 55.49 | 252.10 | 298.52 | 148.5 | 20.87 | 0.53 | 0.602 |
| pH$_{min}$ | 7.97 | 0.11 | 7.52 | 0.11 | 7.20 | 0.08 | 47.91 | **0.000** |
| pH$_{max}$ | 8.43 | 0.35 | 8.16 | 0.15 | 8.15 | 0.37 | 1.04 | 0.390 |
| NH$_3$ [mgL$^{-1}$] | 0.20 | 0.31 | 0.32 | 0.36 | 0.95 | 0.81 | 2.09 | 0.179 |
| NO$_3^-$ [mgL$^{-1}$] | 0.60 | 0.20 | 2.11 | 0.93 | 2.25 | 0.92 | 4.16 | 0.052 |
| NO$_2^-$ [mgL$^{-1}$] | 0.02 | 0.01 | 0.10 | 0.12 | 0.17 | 0.13 | 1.45 | 0.284 |
| Total N [mgL$^{-1}$] | 0.97 | 0.75 | 3.43 | 1.78 | 4.17 | 2.09 | 3.12 | 0.093 |
| PO$_4^{3-}$ [mgL$^{-1}$] | 0.03 | 0.04 | 0.09 | 0.05 | 0.06 | 0.02 | 2.08 | 0.180 |
| **Habitat** | | | | | | | | |
| Flow [m$^3$s$^{-1}$] | 0.83 | 0.55 | 0.45 | 0.39 | 0.44 | 0.32 | 38.06 | **0.000** |
| Depth [m] | 0.29 | 0.14 | 0.54 | 0.34 | 0.50 | 0.33 | 25.89 | **0.000** |
| Substrate index | 22.31 | 5.60 | 7.07 | 5.58 | 6.39 | 3.85 | 422.95 | **0.000** |

**2.2 Macroinvertebrate sampling**
Benthic invertebrate samples were collected in two seasons: autumn (October, 2017) and spring (April, 2018). No flood waves
occurred between these surveys, and the channel morphology remained the same throughout the sampling period. We collected
20 subsamples (1 m$^2$ each subsample) from each low-flow channel along a representative 100 m section of each river according
to the sampling procedure for the BMWP_PL index (Bis and Mikulec, 2013). A total of 480 subsamples were taken from a
wide range of water depths and flow velocity. Following Jowett et al. (1991) and Muñoz-Mas et al. (2016), the substrate types
were converted to a single index by summing the weighted percentages of each type.
Macroinvertebrate samples were collected with a D-frame net according to the Environmental Agency's sampling protocol for
biomonitoring assessment using a kicking motion for 3 minutes across all habitats (Bis and Mikulec, 2013). All collected
material was preserved in the field with 4% formaldehyde. Aquatic macroinvertebrates were separated from the rest of the
material in the laboratory using a stereoscopic microscope and then, they were identified to the family level (Tachet et al.,
2000), except Oligochaeta, Porifera, and Hydrozoa, which were recorded as such. Due to the varied preferences of
macroinvertebrates to habitat conditions, the BMWP_PL index was adopted as the best qualitative index. The Biological
Monitoring Working Party (BMWP) is one of the most commonly used biotic indices in various rivers and streams around the
world (Roche et al., 2010; Wyżga et al., 2013). It has been adopted in many countries, including Poland (Dz.U. 2019 poz.
2149, 2019). The BMWP index was originally developed to represent water quality, but subsequent studies showed that it
reflects ecological quality of the waterbodies and can be also related to hydromorphological impoverishment such like incision
or straightening (Mutz et al., 2013; Wyżga et al., 2013; Mikuś et al., 2021). This index best considers the sensitivity of
invertebrates to environmental variables, because families with similar stress tolerances are grouped together (Armitage et al.,

139 1983).

**2.3 Data analysis**
ANOVA was used to verify the statistical significance of the differences in environmental data between the three river groups
groups (Statsoft, 2013). Non-metric multidimensional scaling (NMDS) was used to test the relationship between the
macroinvertebrate taxonomic composition of the assemblages of the 12 rivers assigned to three groups (Group 1, Group 2 and
Group 3) and hydromorphological variables (water velocity and depth) during the spring and autumn. Descriptive physical
properties (water depth and velocity) were classified into two or three categories: Low, Medium and High. We used minimum
and maximum values of depth and velocity range in each river group and divided them into 33 percentile ranges of the total
value variability. In the case when the ranges were less than 0.5 m depth we have chosen two groups of 50 percentiles of the
depth ranges. The significance of differences between depth and velocity classes was tested by ANOSIM (p-values of pairwise
comparison with Bonferroni correction) on the Bray-Curtis dissimilarity matrix with 499 permutations of the data. PAST
software (version 3.13) was used to analyse NMDS and ANOSIM (Hammer et al., 2001).
To develop habitat suitability functions of macroinvertebrates, reflecting the optimal conditions in the river, generalized
additive models (GAMs) procedures were chosen. The advantage of the method described by Jovett and Davey (2007), is that
it calculates the probability of relations between dependent biotic variables and independent flow parameters. To choose the
best-fitting model, we have ranked the available models according to Akaike information criteria procedure and ΔAICc values,
which reflects the difference of AIC between a given model and the lowest AIC. The best fitting model, describing the
relationship between independent variables (depth and velocity and its two-way interaction between them) and
macroinvertebrate BMWP_PL index, was generalized additive model with Poisson error distribution and log link function.
We have also measured the accuracy of the GAM procedures (Shearer et al., 2015). The total deviance explained calculated
as the relative difference between the residual and the null deviances of the model ([null deviance-residual deviance]/null
deviance) was adopted. The course of the regression line of the BMWP-PL and depth and velocity for each group of the bed
material rivers was obtained using CurveExpert software, where the best fitted line for the set of nonlinear curves was applied
and ranked. The BMWP_PL curve maximum values were regarded as the most optimal for invertebrates and the most
preferred. We were interested in calculation of optimal condition for depth and velocity separately to obtain the optimal
conditions allowing to calculate the discharge which are needed for hydraulic and CCHE2D modelling. The preferred depths
and velocities for each season and river bed material groups were used to calculate the hydraulic discharges which are the most
optimal for BMWP_PL variables and recognized as environmental flow.
**2.4 Hydraulic modelling**
We used the hydraulic method for the assessment of the environmental flow of each river because of the relationship between
the hydraulic parameters of watercourses (depth and velocity) and the quality of the aquatic environment (BMWP_PL - GAM
relations). We used rating curves for each river describing the water depth – flow relations to obtain environmental flow for
given optimal depth. Detailed description of the applied hydraulic method of environmental flow calculation is given in
Książek et al. (2019). To compare the environmental flow in relation to hydromorphological parameters (incision,
redeposition), we used the proportion of Environmental flow ($Q_{env}$) to mean hydraulic parameters of the minimum discharge:
Low Low Flow (LLF- the lowest low flow), Mean Low Flow (MLF – average of the minimum annual flows), and Mean
Annual Flow (MAF – average of the annual flows). Those metrics show the position of the calculated environmental flow in
relation to available water volume (flow characteristics from hydrological year-to-year 1961 to 2017 observations).
**2.5 Case study 2D modelling methodology**
We provided the detailed modelling of a randomly chosen (simple randomization procedure based on the single sequence of
random assignment throwing a dice) one incised and one redeposited river based on CCHE2D model. The model is a depth-
averaged two-dimensional numerical model for simulating unsteady, turbulent, free-surface flow in open channels with a
moveable bed. The CCHE2D model solves depth-integrated shallow water equations for all hydraulic calculations (Wu et al.,
2000; Duan et al., 2001). The CCHE2D package consists of two modules: a Mesh Generator (MG) and a Graphical User
Interface (GUI). The main function of the MG is designing a complex mesh system. The mesh is generated based on the
surveyed topography and/or Digital Terrain Model (DTM). The model was applied in two representative rivers, varying in
river bed morphology – from incised bed rock channels to a channel with natural sediment structures (with redeposition). The
mesh for each sector of the river was generated by interpolating cross sections. A total of 5,112 observations were used: Raba
– 3033 (incision) and Ropa – 2079 (redeposition). The shape of the channels was fairly regular along the reach under study,
and its pattern presented little complexity (i.e., a single channel with no islands), but riffle-pool sequences were observed. The
153–200 m long meshes were composed of cells and nodes (length and number of modes, respectively, for Ropa 153 m, 49715
and Raba 200 m, 99200). Data used for the initial conditions was extracted from field measurements. Special attention was
devoted to bed roughness due to its importance for water surface level. Roughness values ranged from 0.01 in hydraulic smooth
bed zones to 0.07 in rough areas. Finally, the model time step was defined at 0.1 s or 0.25 s, depending on the model structure.
The model was calibrated by comparing the measured and computed water surface levels for measured discharges in all cells
and nodes (Fig. 2).

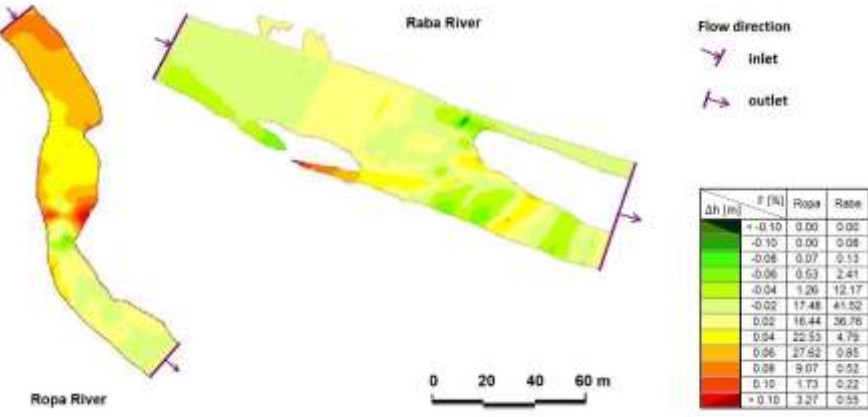


Figure 2. Comparison of calculated and measured water surface levels: The Ropa River for discharge 6.71 $m^3s^{-1}$, and The Raba
River for discharge 10.29 $m^3s^{-1}$ (Δh –difference between measured and calculated water surface level, F - area of particular
differences, percentage).
In the case of the Raba River, for 70% of the calculated nodes, the difference between the calculated and measured water
surface level (WSL) was in the range ±0.02 m. 84% of Ropa River nodes were in the range between of –0.02 to 0.06 m. In all
described models, Δh in the main channel does not cross ±0.02 m, but the visible differences are related to the horizontal layout
of WSL in cross-section. Evaluation of the compatibility measures of the numerical model showed very good accordance
(Książek et al., 2010) and the prepared models did not need recalibration.
For each research section, we choose 20 points at each subsampled area differing in water velocity and water depth as the main
environmental variables creating habitat heterogeneity for macroinvertebrates. Then, according to the relationship between
hydromorphological habitat attributes (water depth and velocity) and the BMWP_PL index values (describing the ecological
quality of the river), we constructed a GAM model as the best fitted method to mark out the range of hydromorphological
attributes (where the BMWP_PL suitability index obtained from the GAM model curve is the highest). Based on the optimal
depth values environmental flow was established using rating curves.
Two rivers (located in the same Carpathian region) representing opposite bed modifications (incision and redeposition) were
chosen for the model as a case study. The modelled sectors of the river had channels with a pool-riffle sequence and fluvial
deposits, but varied in terms of degradation of the bed structure. The hydrological characteristics of the modelled river are
presented in Fig. 3.

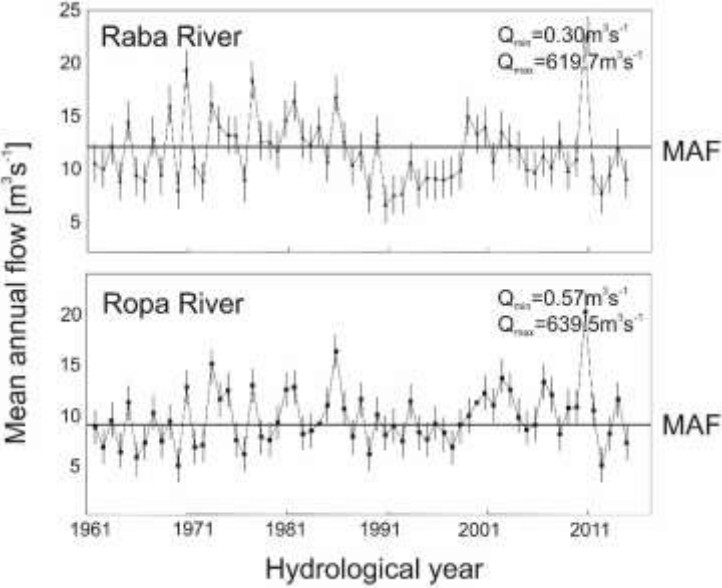


Figure 3. Changes in hydrological regime of the Raba and Ropa Rivers. The horizontal line indicates the Mean Annual Flow
(MAF).

The Raba was selected to represent incised channel rivers (bottom material mainly gravel and small stones, substrate index
14.9). The Dobczyce retention reservoir, which influences the hydrology and morphology of the river, is located upstream of
the examined sector of the river (12 km). Constructing of the retention reservoir in 1986 led to a significant decline in average
annual flow values (MAF values varied from 12.22 $m^3/s$; in 1951-1985 to 10.57 $m^3/s$ in 1986-2015, F = 49.90, p < 0.0001)
and broke the continuity of the sediment transport. The reduction in flow, blockade of sediment supply and longitudinal training
work of the Raba led to incision of the riverbed and permanent compactness of the bed material. The Ropa River, chosen to
represent the redeposition processes, was located among upland, carbonate, and silicate rivers, with the lowest human impact–
agricultural land. The bottom material consists mainly of gravel and sand (substrate index 7.2), where bedload transport
remains undisturbed.
We also wanted to estimate minimum flow values for two rivers which were modelled using CCHE2D. The values of depth
and velocity corresponding to the highest BMWP_PL, obtained from the GAM model for each group of river and season were
plotted against the number of pixels having optimal values. Giving those calculations we were able to obtain the Weighted
Usable Area of macroinvertebrate communities (WUA) showing the most optimal habitat parameters (GAM depth and GAM
velocity). WUA is often defined as an index to various ecological parameters at different organization levels: population (such
as biomass, microhabitat area, size classes) (Muñoz-Mas et al., 2016) or other community level (diversity indices or ecological
metrics) (Jowet, 1997; Jowet, 2003; Theodoropoulos et al., 2015; Pander et al., 2019). Each pixel covered 0.25 $m^2$ of total river
area, so the number of counted calculated cells were the given values of velocity and depth of each group of river were
summarized and multiplied by the surface area. Based on those calculations using CCHE2D model we were able to find the
relationship between usable area and flow values. To calculate the optimal environmental flow values, the curve between flow
and optimal area was created. The low border of optimum of environmental flow was estimated as 50% of WUA values (Jowett
et al., 2008) for CCHE2D modelled rivers.
A hydraulic habitat 2D model of each river section was used for spring and autumn as an example to estimate habitat prediction
in terms of calculated environmental flow during the season. Environmental flow that did not meet the conditions of 100%
habitat suitability for macroinvertebrates was expressed as the critical instream environmental flow value ($Q_{env}$ critical), below
which the parameters of aquatic macroinvertebrate communities dramatically declined.

## 3. Results

### 3.1 Environmental flow based on benthic invertebrates distribution in relation to river hydromorphology

A total of 151 466 individuals belonging to 92 benthic invertebrate families from 480 macroinvertebrate assemblages were identified. High variation was shown in the taxonomic composition of aquatic invertebrates depending on the hydromorphological parameters (water depth and velocity) and the season (Fig. 4). In the case of rivers classified as Group 1, water velocity was found to significantly affect the taxonomic composition of the macroinvertebrates in both spring and autumn (Table 2).

Table 2 Results of ANOSIM analysis comparing macroinvertebrate assemblages between classes of velocity and depth measured for three river groups in the spring and autumn season.

| | | Velocity | | | Depth | | |
|---|---|---|---|---|---|---|---|
| | | Low - Medium | Medium - High | High - Low | Low - Medium | Medium - High | High - Low |
| Spring | Group 1 | **0.1**[*] | **0.21**[**] | **0.1**[*] | -0.01 | | |
| | Group 2 | **0.09**[**] | **0.09**[**] | **0.16**[***] | -0.01 | **0.13**[**] | **0.16**[***] |
| | Group 3 | **0.07**[*] | 0.01 | **0.12**[**] | 0.02 | **0.26**[***] | **0.08**[*] |
| Autumn | Group 1 | 0.04 | 0.06 | **0.09**[*] | 0.0001 | | |
| | Group 2 | **0.15**[***] | **0.25**[***] | **0.39**[***] | **0.07**[*] | **0.3**[***] | **0.13**[***] |
| | Group 3 | 0.04 | 0.03 | 0.03 | 0.03 | **0.11**[**] | 0.01 |

Significance level (p with Bonferroni correction): $*p<0.05$, $** p<0.01$, $*** p<0.001$

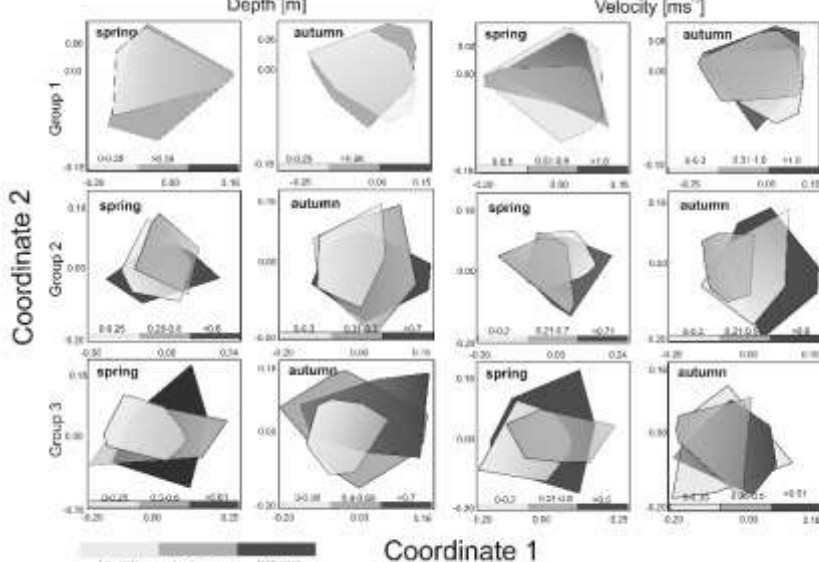

Figure 4. Non-metric multidimensional scaling (NMDS) of macroinvertebrates taxonomic composition of three groups of rivers in the spring and autumn season according to velocity and depth ranges.

In spring, there were significant differences between velocity classes (low and high and medium and high), while in autumn, before overwintering, significant differences were only noted for medium and high classes. In neither season, the differences

noted in taxonomic composition depending on the range of depth were statistically significant in the case of rivers of the second
abiotic group (Group 2), more significant differences were observed between velocity and depth classes (three depth classes
were adopted due to the greater amplitude of these parameters). In the spring, significant differences were visible in all velocity
classes, while in the case of depth they were noted only in the comparison of the low and middle depth classes. In autumn,
differences were found for all classes in the case of variation in both velocity and depth. In the case of Group 3 rivers (carbonate
and silicate fine sediments and rocks), the velocity parameter taxonomically differentiated macroinvertebrate communities
only in the spring between the high and medium velocity classes. In the case of depth, differences were observed in both
seasons – in spring between the deepest and shallowest environments and those with medium depth, and in autumn only
between the deepest and the shallowest zones (Table 2).
Each of the hydromorphological parameters was evaluated by the GAM model, which provided the best fit to the data (Table
3). There were significant effects of depth and velocity and its combination on variation of BMWP_PL index. Generally, the
percentage of the total deviance was the highest for the combination of both hydrological parameters, however depth parameter
alone described similar level of the total deviance. Velocity explained 38.1 and 44.5 % of the total deviance of BMWP_PL
variation in the mountain rivers (Group 1) for spring and autumn respectively. In other rover groups the total deviance
described for velocity varied between 6 to 29 %. Bringing into consideration that both hydrological parameters alone described
more of the total deviance, we regarded them in further analyses separately. The curves of the generalized additive models for
the biotic index BMWP_PL in spring and autumn are presented in Fig. 5.

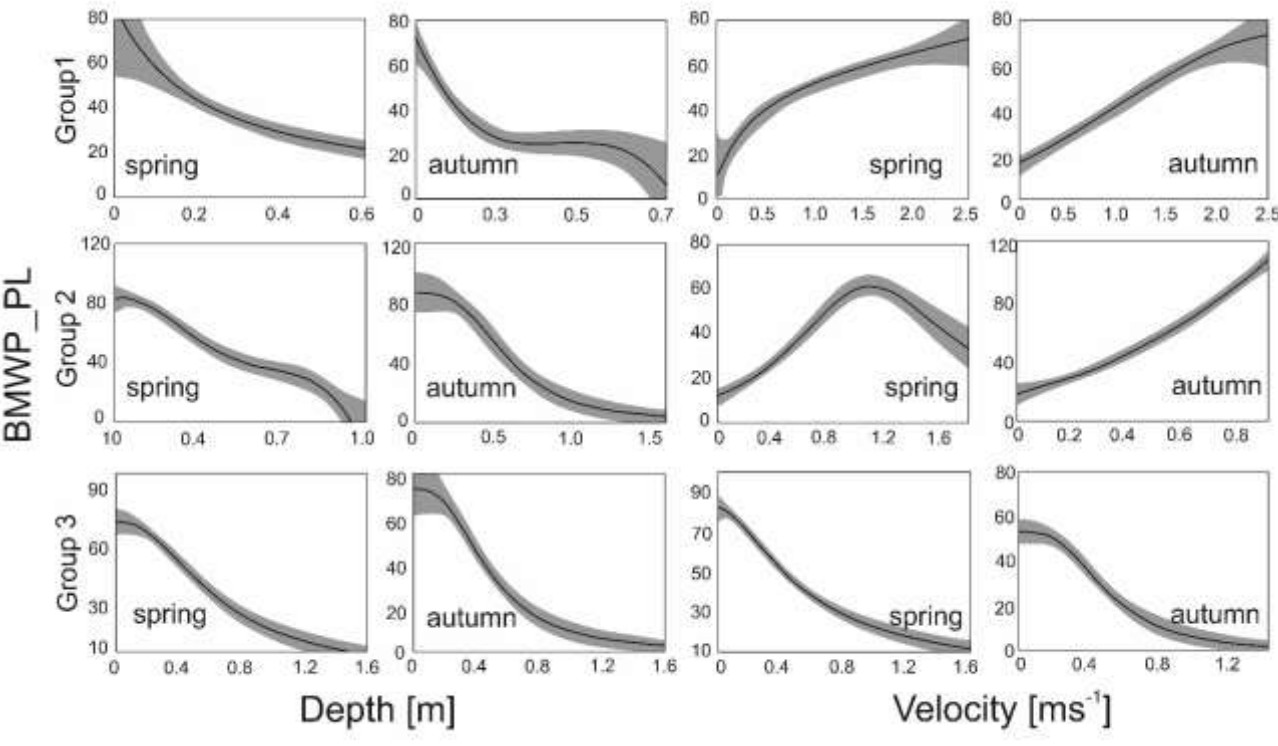


Figure 5. Optimal habitat curves using Generalized Additive Models of BMWP_PL index for water velocity and depth in
spring and autumn season for three river groups.

These models were made for each of the three river groups: calcareous and silica bedrock alpine rivers (Group 1), sandstone
mountain rivers (Group 2), and carbonate and silicate upland rivers (Group 3). In the first group, with a gravel bottom, the
BMWP_PL index reached its highest values at high water velocity and in shallower zones (by the shores). In the second group
of river, the BMWP_PL index was highest at medium velocities in spring and at high velocities in autumn. In both seasons,
higher values for the biotic index were associated with shelf environments, as in the case of Group 1. Similar relationships
with depth were noted in the Group 3 rivers, where BMWP_PL values were highest in the shallow environments at low velocity
in both spring and autumn (Fig. 5).
Table 3 Summary of the Generalized Additive Models for BMWP_PL according to velocity and water depth parameters in
three river groups for spring and autumn season. Res. dev. – residual deviance; % deviance – percentage of total deviance;
Res. df. – residual degrees of freedom; p – significance value.

|  |  | Spring | | | Autumn | | |
|  |  | Group 1 | Group 2 | Group 3 | Group 1 | Group 2 | Group 3 |
|---|---|---|---|---|---|---|---|
| Null | Res.dev. | 2676 | 1324 | 2334 | 2717 | 2632 | 1971 |
|  | % deviance explained | 0 | 0 | 0 | 0 | 0 | 0 |
|  | Res. d.f. | 99 | 99 | 99 | 99 | 99 | 99 |
|  | F | - | - | - | - | - | - |
|  | p | - | - | - | - | - | - |
| Velocity [ms$^{-1}$] | Res.dev. | 1655 | 1250 | 2031 | 1508 | 1890 | 1570 |
|  | % deviance explained | 38.1 | 6.6 | 12.9 | 44.5 | 28.2 | 20.3 |
|  | Res. d.f. | 97 | 96.9 | 96.9 | 97 | 96.9 | 96.9 |
|  | F | 30.66 | 3.01 | 7.9 | 41.46 | 18.41 | 12.1 |
|  | p | <0.0001 | 0.005 | 0.0005 | <0.0001 | <0.0001 | <0.0001 |
| Depth [m] | Res.dev. | 1098 | 762 | 1879 | 1231 | 979 | 1467 |
|  | % deviance explained | 58.9 | 42.4 | 19.4 | 54.6 | 62.7 | 25.5 |
|  | Res. d.f. | 97 | 96.9 | 96.9 | 97 | 97 | 97 |
|  | F | 73.3 | 36.86 | 13.11 | 64.93 | 78.6 | 17.15 |
|  | p | <0.0001 | <0.0001 | <0.0001 | <0.0001 | <0.0001 | <0.0001 |
| Velocity [ms-1] x Depth [m] | Res.dev. | 979 | 672 | 1781 | 1007 | 858 | 1284 |
|  | % deviance explained | 63.4 | 49.2 | 23.6 | 62.9 | 67.4 | 34.8 |
|  | Res. d.f. | 95 | 94.9 | 95 | 94.9 | 94.9 | 95 |
|  | F | 43.41 | 23.63 | 8.45 | 45.04 | 49.2 | 13.48 |
|  | p | <0.0001 | <0.0001 | <0.0001 | <0.0001 | <0.0001 | <0.0001 |


Using the optimal depth characteristics reflecting the habitat suitability (Fig. 5), the environmental flow based on hydraulic
method (rating curve) was defined . The results are shown in Table 4.
Table 4 Environmental flow and flow proportion (S) in different abiotic and bed modification types (I- incision, R-
redeposition) of 12 mountainous rivers.

| River name | Ab. type | River bed mod. | Environmental flow ($Q_{env}$) [m$^3$s$^{-1}$] | | Hydrological characteristics [m$^3$s$^{-1}$] | | | Environmental flow proportion (S) | | | | | |
|  |  |  | spring | autumn | LLF | MLF | MAF | SLLF spring | SLLF autumn | SMLF spring | SMLF autumn | SMAF spring | SMAF autumn |
|---|---|---|---|---|---|---|---|---|---|---|---|---|---|
| Biały Dunajec | I | I | 0.89 | 1.10 | 0.22 | 0.54 | 2.26 | 4.02 | 4.97 | 1.66 | 2.05 | 0.39 | 0.49 |
| Dunajec | I | R | 0.64 | 0.86 | 0.19 | 0.68 | 3.09 | 3.43 | 4.62 | 0.94 | 1.27 | 0.21 | 0.28 |
| Białka | I | R | 0.78 | 0.98 | 0.27 | 0.65 | 3.88 | 2.90 | 3.64 | 1.20 | 1.51 | 0.20 | 0.25 |
| Brynica | II | I | 0.17 | 0.10 | 0.02 | 0.13 | 0.77 | 6.89 | 4.05 | 1.34 | 0.79 | 0.22 | 0.13 |
| Raba | II | I | 4.80 | 3.60 | 0.30 | 3.53 | 11.45 | 16.00 | 12.00 | 1.36 | 1.02 | 0.42 | 0.31 |

| | | | | | | | | | | | | | |
|---|---|---|---|---|---|---|---|---|---|---|---|---|---|
| Toszecki Potok | II | I | 0.27 | 0.18 | 0.02 | 0.11 | 0.59 | 14.35 | 9.56 | 2.43 | 1.62 | 0.46 | 0.30 |
| Biała | II | I | 1.20 | 1.05 | 0.31 | 0.96 | 2.69 | 3.89 | 3.41 | 1.25 | 1.09 | 0.45 | 0.39 |
| Nysa Kłodzka | II | I | 1.90 | 1.50 | 0.14 | 0.61 | 3.68 | 13.37 | 10.55 | 3.12 | 2.46 | 0.52 | 0.41 |
| Sołokija | II | R | 0.36 | 0.50 | 0.25 | 0.72 | 1.34 | 1.42 | 1.97 | 0.50 | 0.69 | 0.27 | 0.37 |
| Warta | III | I | 1.75 | 1.65 | 0.22 | 0.96 | 2.07 | 8.09 | 7.63 | 1.83 | 1.73 | 0.85 | 0.80 |
| Odra | III | R | 7.40 | 7.00 | 4.22 | 9.54 | 42.26 | 1.75 | 1.66 | 0.78 | 0.73 | 0.18 | 0.17 |
| Ropa | III | R | 2.15 | 2.00 | 0.58 | 1.79 | 9.64 | 3.73 | 3.47 | 1.20 | 1.12 | 0.22 | 0.21 |

LLF- Low Low Flow, MLF- Mean Low Flow, MAF- Mean Annual Flow

There is a high variation of the $Q_{env}$, related to its own channel properties and volume of water. To obtain the relation to
hydraulic river parameters, the mean $Q_{env}$ relative similarity to MAF, MLF, and LLF were measured. There was no relation
to the abiotic group of river (Table 5). The only significant relation was linked to channel modification (Fig. 6). In all cases,
the relative similarity of flow was significantly higher in incised channels than redeposited ones.

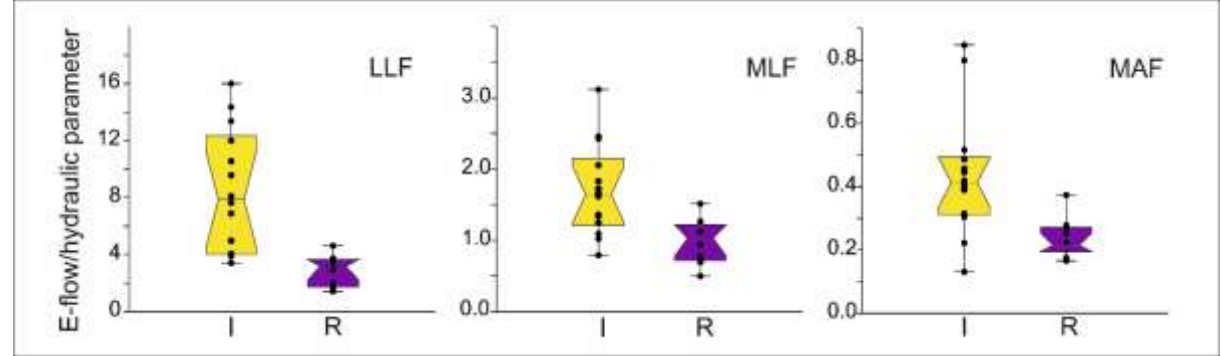


Figure 6.The distribution of mean values ± SE (bix) and whisker length (one sigma) with distribution of jitter of e-flow
proportion to Low Low Flow (LLF), Mean Low Flow (MLF), and Mean Annual Flow (MAF) in relation to river bed
modification (I – incision, R – redeposition).

In each type of flow (MAF, MLF, LLF), the relative similarity was higher in incised rivers than redeposited, showing that the
incised rivers needed much more volume of water to sustain appropriate conditions for macroinvertebrates compared with the
redeposited ones. More detailed analysis and visualization of spatial modelling were predicted by 2D modelling of randomly
chosen rivers presented below as a case study.

Table 5 General linear modelling results for hydrological flow similarity (S) in relation to bed modification (incision and
redeposition), season, and abiotic river group, SS – sum of squares; d.f. – degrees of freedom; MS – mean square.

| Parameter | SS | d.f. | MS | F | p |
|---|---|---|---|---|---|
| LLF$_{sim}$ | | | | | |
| Intercept | 648.66 | 1 | 648.66 | 54.09 | 0.00 |
| **Incision** | **101.13** | **1** | **101.13** | **8.43** | **0.01** |
| Group | 11.28 | 2 | 5.64 | 0.47 | 0.63 |
| Season | 6.32 | 1 | 6.32 | 0.53 | 0.48 |
| Error | 227.86 | 19 | 11.99 | | |
| MLF$_{sim}$ | | | | | |
| Intercept | 41.19 | 1 | 41.19 | 138.07 | 0.00 |
| **Incision** | **3.14** | **1** | **3.14** | **10.52** | **0.00** |
| Group | 0.50 | 2 | 0.25 | 0.84 | 0.45 |
| Season | 0.10 | 1 | 0.10 | 0.33 | 0.57 |

| | | | | | |
|---|---|---|---|---|---|
| Error | 5.67 | 19 | 0.30 | | |
| **MAF$_{sim}$** | | | | | |
| Intercept | 2.70 | 1 | 2.70 | 126.31 | 0.00 |
| **Incision** | **0.32** | **1** | **0.32** | **15.04** | **0.00** |
| Group | 0.11 | 2 | 0.06 | 2.60 | 0.10 |
| Season | 0.00 | 1 | 0.00 | 0.14 | 0.71 |
| Error | 0.41 | 19 | 0.02 | | |

LLF$_{sim}$ – Low Low Flow similarity, MLF$_{sim}$ – Mean Low Flow similarity , MAF$_{sim}$ – Mean Annual Flow similarity

## 3.2 Case study

We calculated the detailed 2D modelling for two randomly chosen incised and redeposited rivers. According to the GAM
macroinvertebrate habitat suitability model, WUA-flow curves were calculated for rivers with varying intensity of bed
modification, Raba (incised) and Ropa (redeposited), as shown in Fig. 7.

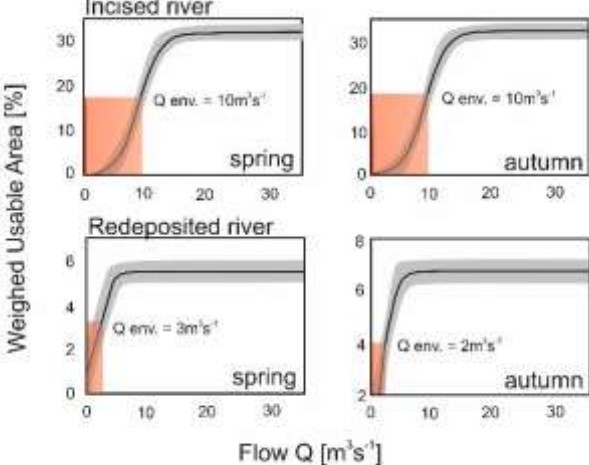


Figure 7. Weighted Usable Area (WUA) - flow relation curves (spring and autumn season) of the rivers varying in bed
modification: Raba River with incision and Ropa River with redeposition.

The environmental flow was defined as the lowest flow corresponding to 50% of the value of the usable area, which ensures
minimum optimal conditions for the development and functioning of aquatic macroinvertebrates (Jowett et al., 2008). Analysis
of the curves for the Raba River shows a 50% reduction in the usable area at the flow of about 10 m$^3$s$^{-1}$ for both spring and
autumn. In the case of the Ropa River, the WUA-flow curves show a 50% reduction in the usable area at the flow of about 2
m$^3$s$^{-1}$ in spring and 3 m$^3$s$^{-1}$ in autumn (Table 6).
A spatial visualization of macroinvertebrate habitat suitability for Q$_{env}$ optimal conditions is presented in Fig. 8. In the case of
the strongly incised Raba River, a very small optimal habitat area was observed, covering only the shelf zone. In the case of
the Ropa River, where sediment transportation occurs, the usable areas constitutes more than 20% of the environmental flow
area. The modelling was also used to determine Q$_{env}$ critical, at which the most valuable areas in terms of habitat (over 80%
suitability) disappear (Fig. 8, Table 6). Below this Q$_{env}$ critical value, a dramatic decline in macroinvertebrate diversity should
be expected.

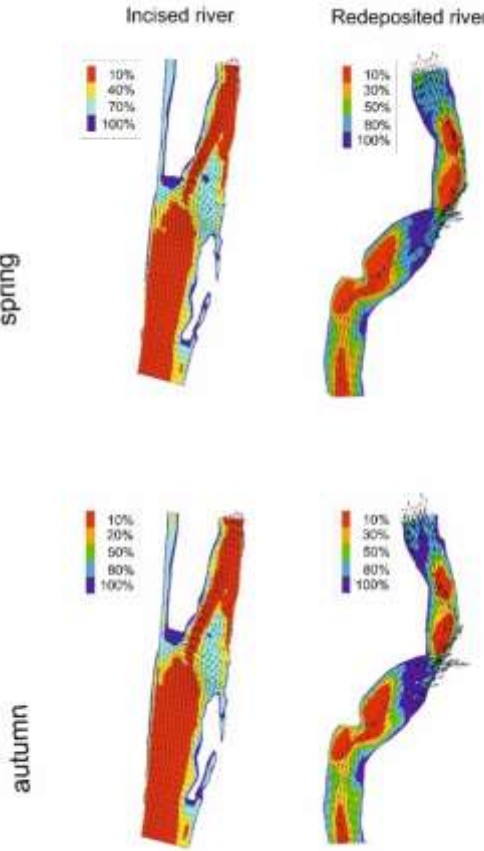


Figure 8. Probability of habitat suitability calculated as a percentage of optimal conditions occurrence of macroinvertebrates

habitat suitability for calculated $Q_{env}$ in spring and autumn season in incised (Raba), and redeposited (Ropa) rivers.

A comparison of the $Q_{env}$ values (optimal and critical) and means: Annual Flow (MAF) and Low Flow (MLF) for the two

types of rivers is presented in Table 7. In the highly incised river (Raba River), the $Q_{env}$ optimal requirement for spring was

lower but for autumn was higher than MAF, and $Q_{env}$ critical was always higher than MLF. In the redeposited Ropa River, in

spring as well as in the autumn season, $Q_{env}$ optimal requirements were much lower than MAF, and MLF was higher than $Q_{env}$

critical. Both findings are congruent with the former hydraulic calculations for all rivers.

349

Table 6 Environmental optimal and critical flow based on macroinvertebrate habitat suitability models of two mountainous

rivers with mean MAF, MLF,  and LLF in relation to the seasons.

| Season | Flow type [$m^3s^{-1}$] | River bed modification | |
|---|---|---|---|
| | | Incision | Redeposition |
| | | Raba | Ropa |
| Spring | $Q_{env\ optimal}$ | 10 | 2 |
| | $Q_{env\ critical}$ | <6 | <1 |
| | MAF | 14.79 | 12.94 |
| | MLF | 5.20 | 2.93 |
| Autumn | $Q_{env\ optimal}$ | 10 | 3 |
| | $Q_{env\ critical}$ | <6 | <1 |
| | MAF | 7.86 | 5.81 |
| | MLF | 3.80 | 1.96 |
| Year | MAF | 11.45 | 9.64 |
| | MLF | 3.53 | 1.79 |
| | LLF | 0.3 | 0.58 |

352

## 4. Discussion

The present study showed that river bed transformation, disturbing sedimentation processes and increasing the incision of the river bed vastly increases the environmental flow values for macroinvertebrates habitat suitability. This is important because incision processes are common in most European rivers (Gore, 1996). Channel incision decreases the area of optimal habitat for macroinvertebrates and increases the potential environmental flow to an extremely high level to obtain the minimum beneficial habitat capacity for macroinvertebrates (Bravard et al., 1997; Skalski et al., 2020). In incised channels, the degree of lateral connectivity between the river and floodplain is reduced, and the degree of modification of the substrate material is higher (Wyżga et al., 2012). As a consequence of channelization and incision, the continuity of the floodplain and shelf zone along the river is disrupted (Walther and Whiles, 2008; Kędzior et al., 2016; Anim et al., 2018; dos Reis Oliveira et al., 2019). Moreover, incision results in a concomitant decrease in sediment supply to the channels, reducing the microhabitat diversity and the quality of macroinvertebrate habitats (Wyżga, 2007; McKenzie et al., 2020). During the incision process, morphological changes in the channel, especially in the case of highly incised rivers, decrease the area of shelf habitat, and fluvial deposits are drastically reduced. Thus, to keep areas wet, flow requirements must be much higher than the mean annual flow and associated with inundation hazards.

Linkage between mean annual flow and environmental flow estimation has been the subject of consideration for many years (Tennant, 1976), based on the assumption that to obtain good stream environment conditions, some percentage of the average flow is required (Richter et al., 2012; Van Niekerk et al., 2019). According to Tennant (1976), 10% of the average flow is the minimum flow recommended to sustain short-term survival habitat for most aquatic life forms. Thirty percent was recommended as a base flow to sustain good survival biota conditions. Sixty percent provides excellent to outstanding habitat for most aquatic life forms during their primary periods of growth and for most recreational uses. However, what about strongly channelized and incised rivers, which are the most common channel types in Europe? Our survey indicated that to obtain high macroinvertebrate diversity, we need a much higher volume of water than 10% of MAF. In the case of incision, a high volume of water is needed to cover the shelves and sediment storage, which are the principal elements of macroinvertebrate habitats and refuges in a dynamic river system (Duan et al., 2009; Anim et al., 2018).

It is obvious that macroinvertebrates are closely linked to the substrate, which is highly variable in terms of particle size (Bravard et al., 1997; Merz and Ochikubo Chan, 2005; Duan et al., 2009). Alluvial processes are strongly disturbed in an incised river, leading to deepening of the channel and bed degradation (Wyżga, 2007). The areas shown in Fig. 7, which are 100% optimal for macroinvertebrates, are extremely narrow in incised rivers throughout the spring and autumn. In most rivers with an augmented bed, the sedimentation process is disturbed, and thus only habitats located closer to the surface, where lateral erosion occurs, provide a optimal habitat for macroinvertebrates. Modern restoration efforts often involve the artificial addition of sediments to sand (dos Reis, Oliveira et al., 2019) or modification of channel morphology to restore the sedimentation process (Violin et al., 2011; Anim et al., 2018).

The biotic integrity of rivers is primarily restricted by downstream transport of sediments controlling the integrity of fluvial ecosystems (Katano et al., 2009; White et al., 2016). Substrate characteristics such as size, stability, compactness, quality, and dynamics are a key parameter determining the occurrence and variation in macroinvertebrate communities. High substrate stability, substrate heterogeneity, and low compactness determine high macroinvertebrate diversity (Beisel et al., 2000; Duan et al., 2009). On the other hand, fine sediments can be regarded as a potential stressor for macroinvertebrates (Meißner et al., 2019). In highly incised sectors of the river, a deficiency of sediment and its compactness as well as a lack of food sources (Shields et al., 1994; Jowett, 2003) lead to impoverishment of the taxonomic composition of macroinvertebrates and favour taxa adapted to high flow only (Wyżga et al., 2013). Our results indicates that prevention of optimal conditions requires more volume of water which exceeds the mean annual flow. This conclusion seems paradoxical and rather dangerous, because increase discharge augments incision processes. We can thus fall into a kind of ecological trap. A solution may be to pay

careful attention to the bed morphology, especially in the case of incised channels. There is still a problem to gather information
on flow ecological response od any organisms and extend the survey in international context should be done (Poff and
Zimmermann, 2010; Fornaroli et al., 2015). We then have two options to preserve the high biodiversity of invertebrates
according to the EU water directive: to vastly increase the water volume or to restore sedimentation processes to obtain a
hydrodynamic balance. As a consequence, optimal habitats for invertebrates and fish will be enlarged. The second option
seems much more realistic. Only then we will be able to successfully maintain the diversity of aquatic biota.

## 5. Conclusions

In habitat modelling, careful attention should be paid to the morphology of the modelled river, its geometry, and the fluvial
processes in the active channel. In incised channels where sedimentation processes are altered, for example, by dam reservoirs
or bedrock downcutting, the area of optimal habitat is limited. Macroinvertebrate habitat preferences are strongly linked to
shelf habitats, where sediment storage and redeposition of bed material is the highest. In that case, the recolonization pattern
of invertebrates requires much higher flows, even higher than the mean annual flow. As a consequence, the river is endangered
by downcutting processes and impoverishment of optimal habitats.
**Author contribution:** Kędzior Renata: Data curation, Formal analysis, Investigation, Resources, Software, Validation,
Writing - review & editing; Kłonowska-Olejnik Małgorzata: Data curation;  Dumnicka Elżbieta: Data curation; Woś
Agnieszka: Investigation; Wyrębek Maciej: Investigation, Resources, Visualization; Książek Leszek: Investigation,
Methodology,  Resources, Validation, Writing - review & editing; Paweł Madej: Funding acquisition, Project administration;
Grela Jerzy: Funding acquisition, Project administration; Skalski Tomasz: Conceptualization, Formal analysis, Investigation,
Methodology, Software, Supervision, Visualization, Writing - original draft.
**Competing interests:** The authors declare that they have no conflict of interest.

## Acknowledgements

The work was partially financed by the National Water Management Authority in Poland "Implementation of the method of
estimating environmental flows in Poland" under the project POIS.02.01.00-00.0016/16 (macroinvertebrates and hydrological
data) and by Ministry of Science and Higher Education in Poland. The statistical analyses used in the paper were supported by
the project "Integrated Program of the University of Agriculture in Krakow" co-financed by European Union under the
European Social Fund. We are grateful to native speaker Sara Wild for improvement of the English text.

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
