# Peer review of "Macroinvertebrate habitat requirements in rivers: overestimation of environmental flow calculations in incised rivers"

_Hydrology and Earth System Sciences, 2021_

## Referee Comment (RC1)

[referee-annotated manuscript omitted]

---

## Author Response (AR1)

Response to Reviewer 1:

Thank you very much for valuable comments, significantly improving the quality of presented paper. Below we tried to include the improvements and detailed comments.

**General Comments:**

The authors test the influence of river incision on environmental flow estimation based on the macroinvertebrate index using data from 12 waterbodies in Poland. The topic is interesting and is one of the emerging trends in environmental and ecological studies. The introductory remarks justify the theme and are well presented.

I suggest to add few sentences to better introduce environmental/minimum flow estimation.

*We added more information in the Introduction: "Environmental Flow is an amount of water required to maintain biological diversity in the river ecosystem. This definition requires to quantify ecological response of aquatic elements to flow alteration, which data are rather scare in the literature (Poff and Zimmerman, 2010). Therefore, it appears crucial to define estimate empirical ranges of environmental flows that ensure optimal habitat conditions for living organisms (Bunn and Arthington, 2002; Acreman et al., 2014)."*

The other background information are up to date and provide sufficient information to put this work in the international context. The sampling campaign is well described in the methods and is suitable for the aim of the study. The statistical analyses should be described in detail to be possible to evaluate their robustness, I would expect a substantial revised paragraph in the next version.

*The paragraph about statistical analysis was improved according to your suggestion.*

Discussion and conclusion sections are generally well written, supported by the results but

Should be compared with the current literature and putted in the international context.

*We added current literature to the discussion:*

[revised manuscript text omitted]

ecological trap. A solution may be to pay careful attention to the bed morphology, especially in the case of incised channels. There is still a problem to gather information on flow- ecological response of any organisms and extend the survey in international context should be done (Poff, and Zimmermann, 2010; Fornaroli et al., 2015). We then have two options to preserve the high biodiversity of invertebrates according to the EU water directive: to vastly increase the water volume or to restore sedimentation processes to obtain a hydrodynamic balance. As a consequence, suitable habitats for invertebrates and fish will be enlarged. The second option seems much more realistic. Only then we will be able to successfully maintain the diversity of aquatic biota.

**Detailed comments:**

*Lines 15-28: the abstract is revised and improved:*

*Prediction of the environmental flow allowing to keep biological diversity and river health developed as a response to the degradation of aquatic ecosystems. Meanwhile the incision and channelization is one of the global common threads, there is still gap in the linking channel incision and environmental flow modelling. The goal of the study was to test the influence of river incision on environmental flow estimation based on the macroinvertebrate communities. The 240 macroinvertebrate assemblages (151 466 macroinvertebrates belonging to 92 families) of 12 waterbodies differing in the bed substrate, amplitude of discharge were surveyed in southern Poland. Generalized additive models supported by nonmetric multidimensional scaling revealed the optimal river parameters (depth and velocity) for the highest values of the biological monitoring working party macroinvertebrate index reflecting the habitat suitability. Using the hydraulic method (rating curve) and 2D modelling the optimal environmental flows for incised and redeposited rivers were estimated. The river incision significantly increased the values of e-flow calculations in relation to redeposited channels. The area of habitat suitability decreased with the bed incision intensity. In incised channels where sedimentation processes are altered by dam reservoirs or bedrock downcutting, the areas of suitable habitat vanish. We conclude that to obtain more suitable conditions covering the shallow zones of the incised river, the higher values of discharge should be applied, up to enormous levels. In habitat modelling, careful attention should be paid to the morphology of the modelled river, its geometry, and the fluvial processes in the active channel.*

Line 41: The subject is missing.

*Improved*

Line 43-45: Substitute "this parameter" with "flow".

*It was improved in the manuscript.*

Line 46: Substitute "demand" with "abstraction".

*It was improved in the manuscript.*

Line 50: Its determination.

*It was improved in the manuscript.*

Line 52: Delete "it".

*It was improved in the manuscript.*

Line 56: This method was introduced much earlier.

*We added the reference of: Gippel C.J., Stewardson M.J. 1998. Use of wetted perimeter in defining minimum environmental flows. Regulated Rivers: Res. Manage. 14, 53-67.*

Line 56: Define "Q" the first time.

*Improved*

*Rewritten the sentences to: Determination of discharge (Q) for environmental flow involves defining the breaking point of the hydraulic variable discharge curves as the e-flow.*

Line 64: Recently the functional flows concept was introduced and should be mentioned here.

*Yes you are right, we added the reference: Yarnell, S. M.; Petts, G. E.; Schmidt, J. C.; Whipple, A. A.; Beller, E. E.; Dahm, C. N.; Goodwin, P.; Viers, J. H. Functional Flows in Modified Riverscapes: Hydrographs, Habitats and Opportunities. Bioscience 2015, 65 (10), 963–972. https://doi.org/10.1093/biosci/biv102.*

*Rewritten the sentences to: In this case, environmental flow is defined in the category of deviation from the natural flow regime (Yarnell et al., 2015).*

Line 65-66: Most of those methods were not specifically developed for macroinvertebrates but, the ones the focus at biological responses were frequently developed for fish. I suggest to mention the most of this methods were developed for fish.

*Improved*

*Rewritten the sentences to: The methods presented above focus on the fish distribution and rarely on diversity and availability of habitats for freshwater macroinvertebrates, which is the most important and sensitive indicator of the ecological state of the ecosystem (Jowett et al., 2008; Birk et al., 2012).*

Line 74: Substitute "array of" with "available in".

*It was improved in the manuscript.*

Line 75: Substitute "vastly" with "greatly", it is not clear which of the previous mentioned methods highlight this need.

*We have delated this sentence because it is highlighted in our results.*

Line 97: Substitute "silica" with "silicate".
*It was improved in the manuscript.*

Line 113: I suggest to include the area of subsamples and the total area sampled for clarity.
*Improved*
*Rewritten the sentences to: We collected 20 subsamples (1 $m^2$ each subsamples) from each low-flow channel along a representative 100 m section of each river according to a sampling procedure for the BMWP_PL index (Bis and Mikulec, 2013).*

Line 122: This choice is not adequately supported, most details should be added.
*Improved.*
*We added sentence: The BMWP index was originally developed to represent water quality, but subsequent studies showed that it reflects ecological quality of the waterbodies and can be also related to hydromorphological impoverishment such like channel incision or straightening (Wyżga et al., 2013; Mutz et al., 2013; Mikuś et al., 2021).*

Line 125-126: This index was developed for water quality, not for e-flow evaluation, this should be stated somewhere as different index (e.g. LIFE) were specifically developed.
*There are two reasons why we have chosen the BMWP_PL index. First of all, in the beginning, when the index was developed it was oriented on pollution biotic index, however in the literature there are information about using BMWP_PL index for the ecological state and habitat quality of the rivers such a depth, velocity, bankfull channel (Wyżga et al. 2013, Mikuś et al. 2021). Mutz et al 2013 also confirmed that Biological Monitoring Working Party score system, used in a gravel bed river responds more to the hydromorphological channel characteristics (either channelized or not) than to water quality.*
*The second reason is related to invertebrate representation in the river. LIFE index is not developed for Poland especially for upland and mountainous gravel bed rivers. The existing in the literature family evaluation LIFE index is not relevant for families in given region. We have collected 128 macroinvertebrate families, and the LIFE scores were proposed only form 75% of the data. As a consequence many sampling scores in spite of high macroinvertebrates diversity had no any LIFE scores. BMWPPL index which is a part of the ecological integrity national monitoring fully covered the 240 assemblages giving more valuable information about diversity loss linked to the e-flow development. Our data confirmed that distribution of BMWPPL had similar pattern as LIFE index (Mantel-Cochran Haenszel test: chi$^2$MH=21.4. p<0.0001).*

Line 164: Those sites are probably "location" within the same "site", please clarify.

*Improved*

*Rewritten the sentences to: For each research section, we choose 20 points at each subsampled area differing in water velocity and water depth as the main environmental variables creating habitat heterogeneity for macroinvertebrates.*

Line 168: "highest" should be better defined here.

*Improved*

*Rewritten the sentences to: we constructed a GAM model as the best fitted method to mark out the range of hydromorphological attributes (where the BMWP_PL suitability index is the highest obtained from the GAM model curve).*

Line 171: Delete "were".

*It was improved in the manuscript.*

Line 181-182: Probably also the sediment load downstream of the reservoir changed substantially unless proper mitigation measures were putted in place. This need to be clearly stated.

*Improved.*

*Rewritten sentence to: The Dobczyce retention reservoir, which influences the hydrology and morphology of the river, is located upstream of the examined sector of the river (12 km). Constructing of the retention reservoir in 1986 led to a significant decline in average annual flow values (MAF values varied from 12.22 m3/s; in 1951-1985 to 10.57 m3/s in 1986-2015, F = 49.90, p < 0.0001) and broke the continuity of the sediment transport. The reduction in flow, blockade of sediment supply and longitudinal training work of the Raba led to incision of the riverbed, change the natural sediment load and permanent compactness of the bed material.*

Line 185: "WUA" should be defined and is still not clear how it was calculated.

*Explained in the text.*

*Rewritten the sentences to: We also wanted to estimate minimum flow values for two rivers which were modelled using CCHE2D. The highest values of depth and velocity obtained from the GAM model (Figure 5) for each type of river and season were plotted against the number of pixels having values resembling suitability model. Giving those calculations we were able to obtain the Weighted Usable Area (WUA) showing the most suitable habitat parameters (GAM depth plus GAM velocity). Each pixel covered 0.25 m² of total river area, so the number of counted calculated cells were the given values of velocity and depth of each type of river were summarized and multiplied by the surface area. Based on those calculations using CCHE2D model we were able to find the relationship between usable area and*

*flow values. To calculate the optimal environmental flow values the curve between flow and suitable area was created. The optimum of environmental flow was estimated as 50% of WUA values for CCHE2D modelled rivers below which the habitat retention level will have negative effect on populations conditions (Jowett et al., 2008).*

Line 196: These categories should be clearly reported.

*Improved.*

*Rewritten the sentences to: Descriptive physical properties (water depth and velocity) were classified into two or three categories: Low, Medium and High. We used minimum and maximum values of depth and velocity range in each river group and divided them into 33 percentile ranges of the total value variability. In the case when the ranges were less than 0.5 m depth we have chosen two groups of 50 percentiles of the depth ranges.*

Line 201-202: The description of the GAM models should be substantially improved as the details provided do not allows for a proper evaluation.

Line 202-204: As for GAM, GLM should be defined and properly described.

*We improved the description of the GAM and GLM as well as we improved all Data Analysis as below: "ANOVA was used to verify the statistical significance of the differences in environmental data between the three river types groups (Statsoft, 2013). Non-metric multidimensional scaling (NMDS) was used to test the relationship between the macroinvertebrate taxonomic composition of the assemblages of the 12 rivers assigned to three types groups (Group 1, Group 2 and Group 3) and hydromorphological variables (water velocity and depth) during the spring and autumn. Descriptive physical properties (water depth and velocity) were classified into two or three categories: Low, Medium and High. We used minimum and maximum values of depth and velocity range in each river group and divided them into 33 percentile ranges of the total value variability. In the case when the ranges were less than 0.5 m depth we have chosen two groups of 50 percentiles of the depth ranges. The significance of differences between depth and velocity classes was tested by ANOSIM on the Bray-Curtis dissimilarity matrix with 499 permutations of the data. PAST software (version 3.13) was used to analyze NMDS and ANOSIM (Hammer et al., 2001).*

*To develop habitat suitability functions of macroinvertebrates, reflecting the optimal conditions in the river, generalized additive models (GAMs) procedures were chosen. The advantage of the method described by Jovett and Davey (2007), is that it calculates the probability of relations between dependent biotic variables and independent flow parameters. To choose the best-fitting model, we have ranked the available models according to Akaike information criteria procedure and ΔAICc values, which reflects the difference of AIC between a given model and the lowest AIC. The best fitting model, describing the relationship between independent variables (depth and velocity and its two-way interaction between*

*them) and macroinvertebrate BMWP_Pl index, was generalized additive model with Poisson error distribution and log link function. We have also measured the accuracy of the GAM procedures (Shearer et al., 2015). The total deviance explained calculated as the relative difference between the residual and the null deviances of the model ([null deviance-residual deviance]/null deviance) was adopted. The course of the regression line of the BMWP-PL and depth and velocity for each group of the bed material rivers was obtained using CurveExpert software, where the best fitted line for the set of nonlinear curves was applied and ranked. The BMWP_PL curve maximum values were regarded as the most suitable for invertebrates and the most preferred. We were interested in calculation of optimal condition for depth and velocity separately to obtain the optimal conditions allowing to calculate the discharge which are needed for hydraulic and 2CEDD modelling. The preferred depths and velocities for each season and river bed material groups were used to calculate the hydraulic discharges which are the most suitable for BMWP_Pl variables and recognized as environmental flow."*

Description of mixed models in results:

*There were significant effects of depth and velocity and its combination on variation of BMWP_PL index. Generally, the percentage of the total deviance was the highest for the combination of both hydrological parameters, however depth parameter alone described similar level of the total deviance. Velocity explained 38.1 and 44.5% of the total deviance of BMWP_PL variation in the mountain rivers (Group 1) for spring and autumn respectively. In other river groups the total deviance described for velocity varied between 6-29%. Bringing into consideration that both hydrological parameters alone described more of the total deviance, we regarded them in further analysis separately.*

Line 217: I suggest to use transparency as it is difficult to understand how much the polygons are overlapping.

*Improved.*

*Revised Fig. 4 is below.*

[Figure]

Line 234: Please, use the same y scale for the different plots.

*We would like to stay on the scales which were defined for each river group range. When we expand the ranges, the models extrapolate the curves to the given extensions. So we would like to avoid the extrapolation of the data.*

Line 238-239: This is not necessarily true, the use of univariate models (there are not enouth details in the methods to evaluate this point) do not account for both velocity and depth at the same time. In this case the author can expect "highest values at high water velocity OR in shallower zones".

See Fornaroli, R.; Cabrini, R.; Sartori, L.; Marazzi, F.; Canobbio, S.; Mezzanotte, V. Optimal Flow for Brown Trout: Habitat – Prey Optimization. Sci. Total Environ. 2016, 566–567, 1568–1578. for an example of multivariate regression.

*It was done purposely to reach the values of depth and velocity separately to include them later in the hydrological and CCHE2D modelling. We agree that much better will be to obtained both factors together in the mix model. But it was 100% methodological purpose.*

Line 242: This point should be clarified.

*Improved.*

*In the first version of the paper, we used several hydraulic methods, the final version of the manuscript was based only on the hydraulics of flows in the rating curve, excluding the wetted periemter method.*

*At the stage of editing, the description of this method was not removed. The text was clarified both in the results and in the methodology.*

*Rewritten sentence to: "Using the optimal depth characteristics reflecting the habitat suitability (Fig. 5), the environmental flow based on rating curve was defined. The results are shown in Table 4."*

Line 251: The table must be read without the text, there are too many acronyms that make difficult to get the content of the table.

*We would like to leave the acronyms of flow type in the table, but we added acronyms explanation under the table for better understanding the meaning of the table 4.*

Line 271: Delete space.

*It was improved in the manuscript.*

Line 276: The lowest flow.

*Improved*

Line 276-278: A relevant citation should be inserted here.

*This sentence was deleted and developed in the discussion chapter.*

Line 282-283: This is unclear, please reword.

*Improved*

Line 285: Substitute "would have to " with "should".

*It was improved in the manuscript.*

Line 301: Delete one dot.

*It was improved in the manuscript.*

Line 304: Delete the first dot.

*It was improved in the manuscript.*

Line 304: Substitute "connectivity of" with "connectivity between".

*It was improved in the manuscript.*

Line 323: Substitute "varied" with "variable".

*It was improved in the manuscript.*

Line 325-326: This sentence need to be reworded.

*Improved*

**Response to Reviewer 2**

We are very grateful for valuable comments, showing imperfection of the text. We believe that detailed explanations and clarification of our aims as well as whole manuscript will fully satisfy the reviewer and readers interested in the topic. Below we present all improvements and detailed responses for each comment.

**General comments**

The authors used macroinvertebrate sampling, environmental measurements, and hydraulic habitat modelling to compare calculated environmental flow requirements between incised rivers of various types and rivers with sediment deposition. The topic is novel and interesting, but the methods are inadequately explained (please see detailed comments below) and sometimes appear inappropriate. In addition, parts of the manuscript are quite difficult to understand.

Areas of particular concern are:

(1) Insufficient background information, such as the nature and causes of the incision and sedimentation.

*Improved. We added more information about incision and sedimentation to the Introduction:*

*"Another parameter, which is usually neglected in flow modelling, is associated with morphological channel modification and incision (Wyżga et al., 2012; Skalski et al., 2016). Incision and channel simplification is a global problem overwhelming most of the rivers in the mountain as well as in lowland areas (Skarpich et al., 2020). During the last 100 years anthropogenic processes related to river regulation related to narrowing and straitening disturbed the fluvial processes leading to enormous river incision (Rinaldi et al., 2005, Wyżga, 2008). As a results rivers become a vertically closed systems loosing the ability to store alluvial material. Moreover incision up to the bedrock simplifies the microhabitat array of the river (Neachell, 2014) and lead to elimination most of the habitats (Muñoz-Mas et al., 2016) as well as affect ecosystem functioning (biodiversity lost and food web network simplification, Shields et al. 1998; Jeffres et al., 2008). To preserve the habitats linked with alluvial floodplains and microhabitats the environmental flow should be vastly increased and full fill the whole river active channel."*

(2) The use of a pollution-oriented biotic index (BMWP) as a biotic response variable rather than a flow-oriented index such as LIFE (Lotic-invertebrate Index for Flow Evaluation).

*There are two reasons why we have chosen the BMWP_PL index. First of all, in the beginning, when the index was developed it was oriented on pollution biotic index, however in the literature there are information abut using BMWWPL index for the ecological state and habitat quality of the rivers such a depth, velocity, bankfull channel (Wyżga et al., 2013 in Hydrobiologia, Mikuś et al., 2021 in Ecological Engineering). Mutz et al. (2013 in Hydrobiologia) also confirmed that Biological Monitoring Working*

*Party score system, when used in a gravel bed river responds more to the channel characteristics (either channelized or not) than to water quality.*

*The second reason is related to invertebrate representation in the river. LIFE index is not developed for Poland, especially for upland and mountainous gravel bed rivers. The existing in the literature family evaluation LIFE index is not relevant for families in given region. We have collected 128 macroinvertebrate families, and the LIFE scores were proposed only form 75% of the data. As a consequence many sampling scores in spite of high macroinvertebrates diversity had no any LIFE scores. BMWPPL index which is a part of the ecological integrity national monitoring fully covered the 240 assemblages giving more valuable information about diversity loss linked to the e-flow development. Our data confirmed that distribution of BMWPPL had similar pattern as a LIFE index (Mantel-Cochran Haenszel test: $chi^2MH=21.4.$ $p<0.0001$).*

(2) Reference to WUA (presumably weighted usable area) without explanation of which species and life-history stage it was calculated for and how.

*In this case WUA was linked to BMWP_PL optimal values not any species or life history.*

*The definition of the Bovee and Cochnauer 1977 (Bovee, K.D., and T. Cochnauer. 1977. Development and evaluation of weighted criteria, probability-of-use curves for instream flow assessments: fisheries. Instream Flow Information Paper 3. United States Fish and Wildlife Service FWS/OBS-77/63. 38pp.) the weighted usable area is defined as a total surface area having a certain combination of hydraulic conditions multiplied by the composite probability of use for that combination of conditions. These procedure equates an area of marginal habitat to equivalent area of optimal habitat. Optimal habitat for macroinvertebrates was obtained from the curves confirmed by Generalized Additive modelling (Figure 5). We can not agree that is linked only to species or a life history stage. In our case optimal habitat for macroinvertebrates is reached when the BMWPPL index is the highest. From this point obtaining the highest of value of depth and velocity we were able to calculate in our case study number of cells which achieve both hydrological conditions (depth and velocity). On that information building the 2D model of the river bed and hydraulic conditions we were able to estimate the probability of presence of optimal areas for invertebrates.*

(4) Incorporation of multiple statistical testing that is likely to increase type I error and does not seem necessary to address the stated study aims.

*Multivariate tests suffer from loss of power and type I error inflation in the presence of heteroscedasticity and sample size imbalances. In our studies all sample points were the same (one square meter). Also the variability of a variable was equal across the range of gradients, so the heterodascity was low. I guess that you are mentioned about multiple comparison test Error type I in ANOSIM analysis. In that case the Bonferroni correctness was applied for p-values to avoid the incidence significance.*

(5) Lack of discussion of limitations of the methods used.

*The only limitation of the method is that we need many empirical data from the filed which is uncommon when environmental flow is calculated. Number of sites with broad gradient of flow alternation increase the sampling procedure for large rivers where the conditions are different than in those presented in the paper. Build the best bio-index which could be developed across large area e.g. Europe or Palearctic and include national and international monitoring assessment agency to develop one multifunctional monitoring program. More explanation in the last paragraph.*

Specific comments referenced by line number(s)

15-29. The abstract is quite poorly written and structured and often difficult to understand.

*The whole abstract was improved as below:*

*"Flow variability determines the conditions of river ecosystem and river ecological functioning. The variability of ecological processes in river ecosystems gradually decreases due to river channelization and incision. Prediction of the environmental flow allowing to keep biological diversity and river health developed as a response to the degradation of aquatic ecosystems overexploited by human. The goal of the study was to test the influence of river incision on environmental flow estimation based on the macroinvertebrate biological monitoring working party macroinvertebrate index. The 240 macroinvertebrate assemblages of 12 waterbodies differing in the bed substrate, amplitude of discharge were surveyed in southern Poland. The variations in the distribution of 151 466 macroinvertebrates belonging to 92 families were analyzed. The similarity of benthic macroinvertebrates reflects the typological division of the rivers into three classes: mountain Tatra streams, mountain flysch rivers, and upland carbonate and silicate rivers. As a response variable reflecting the macroinvertebrate distribution in the river, environmental parameters, BMWP_PL index was chosen. The river incision significantly increased the values of e-flow calculations in relation to redeposited channels. The area of habitat suitability decreased with the bed incision intensity. In highly incised rivers, the environmental flow values are close to the mean annual flow, suggesting that a high volume of water is needed to obtain good macroinvertebrate conditions. As a consequence, the river downcutting processes and impoverishment of suitable habitats will proceed."*

Which measurable variable(s) does "water flow intensity" refer to? Discharge? Velocity? Stream power?

*Discharge, rewritten in the text.*

There is no need to include "multispecies". By definition, and ecological community comprises multiple species.

*Improved.*

The morphology of what?

*River channel morphology - rewritten in the text*

What is an "incision dam"?

*Improved, we added the dot between incision and dams.*

Perhaps "characteristics" rather than "values".

*Improved, we used "characteristics".*

56-57. Q should be defined and a citation should be provided for the use of discharge curves.

*Rewtitten the text to: „Determination of Q discharge value (Q) for environmental flow involves defining the breaking point of the hydraulic variable discharge curves as the e-flow (Gippel and Sterwardson 1998, Vezza et all.2011, Tare and all. 2017)"*

The methods referred to have also been applied to other organisms, particularly fish.

*Yes, also fish. But it can be applied for benthos, algi, microbes and fungi too. See Poff and Zimmerman, 2010 (Ecological responses to altered flow regimes: a literature review to inform the science and management of environmental flows, Freshwater Biology, 55, 2010).*

76-77. I cannot find anything in the manuscript that considers macroinvertebrate habitat preferences. Habitat preference varies among species and their life-history stages, and the manuscript does not consider individual species and stages.

*BMWP_PL index values refer to habitat preferences of whole community, not only particular species or life history. See Fig. 4 and 5. ( according to the results: Wyżga et al., 2013).*

*Rewritten sentence to: The goal of the study was to test the influence of river incision on environmental flow estimation based on the macroinvertebrate biological monitoring working party macroinvertebrate index.*

80-82. The intended meaning of "to identify a scale of e-flow overestimation" and "overestimation of e-flow calculations" is not clear to me.

*Rewritten to: identify a reality of providing e-flow value for different hydromorphological modifications in relation to available amount of water*

*Our empirical data indicate that to obtain the sufficient flow which is optimal for macroinvertebrates in incised rivers, we need a gross amount of water. Otherwise if we will stop on low flow levels, the invertebrate population will be declined.*

*To preserve the habitats linked with alluvial floodplains and microhabitats the environmental flow should be vastly increased and full fill the whole river active channel. See introduction concerning incision.*

Why did you sample only mountain rivers when your objectives (line 78) refer to mountain and lowland rivers? Also, why did you select these particular river types and sites?

*Our survey was provided in three classes of rivers from typical gravel bed mountain river, throughout mountainous sandstone and silica rivers to upland landforms with various carbonate and silicate sediments characteristic for lowland rivers. We have chosen those rivers to obtain similar regional values. We wanted to have the data from the same flow conditions so we needed to collect invertebrates from relative period of time otherwise if the river will be far enough the survey would be longer and probability of changes of flow conditions will be higher. So we will need to add a date as a covariable, then we decided to focused on limited area (but still wide) providing high variation in bed classes not autocorrelated in space. We also consulted the choice of the rivers with the Polish Water Agency to have similar water quality conditions.*

97-109. The river categories are variously referred to as classes, groups and types. It would be better to stick to a consistent term.

*Improved in the text.*

*Classes refers to the bed type according to the Polish Water National Authority and the Water Framework Directive. Types were created according to the type of channel modification: incision and redeposition.*

Some detail about the incision in these rivers would be informative, for example the spatial extent and rates of incision and deposition and the extent to which these processes are natural or induced by anthropogenic changes in land cover, land use and flow regimes.

*Improved: The typology of river channel modification was obtained from field observation and channel measurements (cross-sections, longitudinal profile and cover, high of the floodplain). Narrow channels with downcutting to the floodplain and simplified channel morphology ware defined as incised.*

Please explain what the substrate index measures and how it is calculated.

*We measured the substrate index according to Muñoz-Mas, R., Papadaki, C., Martínez-Capel, F., Zogaris, S., Ntoanidis, L., and Dimitriou, E.: Generalized additive and fuzzy models in environmental flow assessment: A comparison employing the West Balkan trout (Salmo farioides; Karaman, 1938), Ecological Engineering, 91, 365-377, https://doi.org/10.1016/j.ecoleng.2016.03.009, 2016: "Substrate composition was converted into a single value through the Substrate index [−], by summing the weighted percentages of each substrate type as follows: Substrate index = 0.08 × Bedrock% + 0.07 × Boulder% + 0.06 × Cobble% + 0.05 ×Gravel% + 0.04 × Fine Gravel% + 0.03 × Sand%."*

I do not understand the reason for using the BMWP index because it is related to pollution rather than flow velocity. Why did you not use a flow-specific index such as LIFE (Lotic-invertebrate Index for Flow Evaluation)?

*We know the LIFE index, however existing classification is useless for mountain and mountainous rivers. In many cases LIFE values was null. We proved that BMWP_PL index is not related only to the pollution but rather to ecological state of the river (see Wyżga et al., 2013, Hydrobiologia, Mikuś et al., 2021, Ecological Engineering).*

Książek et al. (2019) is in Polish and therefore will not be accessible to most potential international readers. You could perhaps refer to Gippel and Stewardson (1998, Regulated Rivers: Research and Management).

*We do not understand. It is in Springer English.... We added also the Gippel and Stewardson (1998) reference to this part of the paper.*

How did you define low flow? What threshold was used and why?

*Low mean flow is the mean of the minimum annual flows from1961-2017, Low Low Flow is the minimum annual flow (1961-2017).*

What procedure did you use to ensure the choice was random.

*From among 12 rivers we selected one which was incised and second which was with the redeposition to obtain a detailed case studies with 2DD models. We used the simple randomization procedure based on the single sequence of random assignment throwing a dice.*

For Fig. 2, please tell the reader what delta h and F represent and how they were calculated.

*Improved, we added to the caption of the figure 2: (Δh –difference between measured and calculated water surface level, F - area of particular differences, percentage).*

For Fig. 3, please explain what the error bars represent.

*Standard errors for set of day flows.*

*The bars represent confidence interval for mean from month mean values and daily variation. We added to the caption of the figure 3: "The horizontal line indicates the mean annual flow (MAF)"*

The acronym WUA (presumably referring to weighted usable area) needs to be explained.

*It was explained in the text.*

How did you calculate weighted usable area? It is normally defined with respect to a particular life-history stage of a particular species, and calculated from its preferences for velocity, depth and substratum.

*Weighted usable area is defined as area of given parameters which is used by any biotic parameter, such as species abundance, assemblage diversity or ecological indices. See general comments.*

What thresholds were used to classify depth and velocity as low, medium and high, and on what basis were the thresholds chosen?

*Please see revised figure 4. There were build the models of BMWP_PL values in two dimensions - depth and velocity. From modelled curves we were able to indicate the optimal condition to higher values of BMWP_PL index.*

*We added more information about classes of velocity and depth: "Descriptive physical properties (water depth and velocity) were classified into two or three categories: Low, Medium and High. We used minimum and maximum values of depth and velocity range in each river group and divided them into 33 percentile ranges of the total value variability. In the case when the ranges were less than 0.5 m depth we have chosen two groups of 50 percentiles of the depth ranges."*

[Figure]

*Revised figure 4*

Did you apply any transformation to raw abundance values before calculating the Bray-Curtis index?
*Our data composed of density of each family per square meter so we didn't have to transform the data to raw form.*

219-229. Text in this paragraph is often hard to follow. In addition, there is statistical problem of multiple testing (32 separate tests in Table 2), which increases the type I error rate. Also, it is very well known that stream invertebrate assemblages vary according to velocity and depth, and unclear why all of these analyses are need to fulfil the study aims.

*We improved table 2, see below:*

| | | Velocity | | | Depth | | |
|---|---|---|---|---|---|---|---|
| | | Low - Medium | Medium - High | High - Low | Low - Medium | Medium - High | High - Low |
| Spring | Group 1 | **0.1**[*] | **0.21**[**] | **0.1**[*] | -0.01 | | |
| | Group 2 | **0.09**[**] | **0.09**[**] | **0.16**[***] | -0.01 | **0.13**[**] | **0.16**[***] |
| | Group 3 | **0.07**[*] | 0.01 | **0.12**[**] | 0.02 | **0.26**[***] | **0.08**[*] |
| Autumn | Group 1 | 0.04 | 0.06 | **0.09**[*] | 0.0001 | | |
| | Group 2 | **0.15**[***] | **0.25**[***] | **0.39**[***] | **0.07**[*] | **0.3**[***] | **0.13**[***] |
| | Group 3 | 0.04 | 0.03 | 0.03 | 0.03 | **0.11**[**] | 0.01 |

*NMDS analyses were obtained to test if invertebrates composition differ in relation to depth and velocity, to be sure that application of one of the invertebrate indices (BMWP_PL) reflects the changes in macroinvertebrate community structure and taxonomic composition. Our results, however detailed, but showed that in each group of rivers the composition is related to depth and velocity. We do know that in multiple comparisons tests the chance of commitment of Type I error increases. To resolve the problem, all p values were obtained with Bonferroni correction.*

232-233. The fact that the relationship of the BMWP index to depth and velocity is so variable further indicates to me that it is not a suitable biological response variable for the purposes of this study.

*You misunderstand the problem. We did not want to evaluate relation of BMWP to velocity and depth but rather BMWP_PL was a measure of good conditions for invertebrate communities. It allows to estimate the frames of depth and velocity optimal conditions.*

How was similarity calculated?

*The rate and shape of obtained curves.*

Please explain what the various symbols (dots, lines, boxes) in Fig. 6 represent.

*It is typical mean values box and whisker (standard error)also with the jitter of each data.*

*We improved the caption of the Figure 6. The distribution of mean values ± SE (box) and whisker length (one sigma) with distribution of jitter of e-flow proportion to LLF, MLF, and MAF in relation to river bed modification. I – incision, R – redeposition.*

281-285. What do the percentages in Fig. 8 represent? Percentages of what?

*Percentage of suitability conditions*

*Rewrtitten: „Figure 8. Probability of habitat suitability calculated as a percentage of optimal conditions occurrence percentage of habitat suitability of macroinvertebrates index for calculated Qenv in spring and autumn season for incised (Raba), and redeposited (Ropa) rivers,,.*

302 and elsewhere. Habitat for what? Different species have different habitat requirements and what is suitable for one species at one life-history stage may be unsuitable for another species and life history stage, or even another life-history stage of the same species.

*Habitat suitability for invertebrates according the BMWP_PL values. We improved it in the text.*

338-339. Because this conclusion seems rather paradoxical, I suggest that you should discuss the limitations of your method. Various types of habitat simulation methods have received significant evaluation and criticism (see for example Gan & MacMahon 1990, Regulated Rivers: Research and Management; Parasiewicz and Walker JD 2007, River Research and Applications; Railsback 2017, Fisheries; Yi et al. 2017, Renewable and Sustainable Energy Reviews).

*To be honest, our goal was not to implement any new method of e-flow modeling, we just wanted to show that we should include in e-flow estimation not only "The Instream Flow Incremental Methodology" but also habitat preferences and channel morphology. We haven't been focusing on simulation methods which are well developed, but the problem is that we still have insufficient set of data which can be included in the simulations. There is still a problem to gather information on flow-alteration and ecological response of any organisms, invertebrate assemblages, plant communities or microbiomes (eg. review of Poff and Zimmerman, 2010, Freshwater Biology). If we agree that environmental flow is an amount of water required to maintain biological diversity in the river ecosystem we should find the modelled habitat preferences of any given diversity group. In our study we used the distribution of 240 invertebrate assemblages in various rivers and streams. The most important element of the survey was to obtain optimal depth and velocity conditions for invertebrates. As a measure of invertebrate ecological success we used the BMWP_PL index, which described well invertebrate diversity. Then using generalized linear modelling we were able to find the curves of BMWP_PL-depth and BMWP_Pl-velocity relation. This suitability method have advantages than conventional habitat suitability criteria and the associated composite suitability index which were described by Jowett and Davey (2007). But the most important is that it can make quantitative*

*predictions of biotic parameter at given flows. Then using rating curves of depth flow relation at each cross section or CCHE2D modelling we were able to estimate the flow which is needed to maintain biological diversity. Those results were compared with hydraulic parameters of each river which are also used in many models and easy to be collected from the water agencies (such like Low Flow or Mean Annual Flow). We found that to obtain optimal conditions for biological diversity we should need high amount of water, even higher tnan mean annual flow. This conclusion seems paradoxical, but apparently show the environmental situation in incised rivers. Our previous findings (Wyżga et al 2009, River Res Appl , Wyżga et al 2012 – Hydrobiologia,, Skalski et al 2015 – River Research an Application, Skalski et al 2020, STOTEN) showed that downstream cutting and incision negatively influence on freshwater macroinvertebrates, fish and riverine invertebrates. Even keeping the flow at high level we cannot obtain conditions sufficient for macroinvertebrates, fish and eg. predatory riverine ground beetles which needs flat shallow microhabitats. The ecosystem becomes closed and impoverished even if we establish e-flow coming from the general model. We then rather focused in this paragraph on signaling the problem and development of international survey program including fish, macroinvertebrate, plant specialist as well as hydrologist, geomorphologist and modelers to reach the results in international context. We are able just to give a sign that we are still at the beginning of our race.*

We propose the version of last paragraph:

*The biotic integrity of rivers is primarily restricted by downstream transport of sediments controlling the integrity of fluvial ecosystems (Katano et al., 2009; White et al., 2016). Substrate characteristics such as size, stability, compactness, quality, and dynamics are a key parameter determining the occurrence and variation in macroinvertebrate communities. High substrate stability, substrate heterogeneity, and low compactness determine high macroinvertebrate diversity (Beisel et al., 2000; Duan et al., 2009). On the other hand, fine sediments can be regarded as a potential stressor for macroinvertebrates (Meißner et al., 2019). In highly incised sectors of the river, a deficiency of sediment and its compactness as well as a lack of food sources (Shields et al., 1994; Jowett, 2003) lead to impoverishment of the taxonomic composition of macroinvertebrates and favour taxa adapted to high flow only (Wyżga et al., 2013). Our results indicates that prevention of optimal conditions requires more volume of water which exceeds the mean annual flow. This conclusion seems paradoxical and rather dangerous, because increased discharge augments incision processes. We can thus fall into a kind of ecological trap. A solution may be to pay careful attention to the bed morphology, especially in the case of incised channels. There is still a problem to gather information on flow- ecological response of any organisms and extend the survey in international context should be done (Poff, and Zimmermann, 2010; Fornaroli et al., 2015). We then have two options to preserve the high biodiversity of invertebrates according to the EU water directive: to vastly increase the water volume or to restore sedimentation processes to obtain a hydrodynamic balance. As a consequence, suitable habitats for invertebrates and fish will be enlarged. The second option seems much more realistic. Only then we will be able to successfully maintain the diversity of aquatic biota.*

---

## Author Response (AR2)

Thank you very much for the manuscript comments, some comments were valuable and we included it in the text but some more general, defining the habitat or weighed usable area cannot be accepted by authors in the light of current publications and environmental flow concept development. Taking from reviewer#2 point of view we should model only fish populations to obtain the e-flow calculations. Many papers however describe the habitat at various levels of biotic organization from individuals to the ecosystems.

Detailed comments to the reviewer#2 are included below:

REF#2: Please see references listed below regarding correct use of the word "habitat" and related terms. Habitat is organism-specific, and defined as the resources and conditions that enable a particular organism to occupy a particular place. It is not a term than can be applied to a community (e.g., revised manuscript, line 85) and the BMWP_PL index is not a measure of "habitat suitability" (lines 152-158).

Indeed, phrases such as "suitable habitat" (e.g., line 28) and "usable habitat" (e.g., line 205) should not be used – the words "suitable" and "usable" are redundant because if an area is not suitable or usable for an organism, it is not part of its habitat.

Authors: This phrase was deleted from the text. We have replaced the term "suitable area" to "optimal area" informing about optimal conditions for invertebrates modelled from GAM rather than of conventional habitat suitability criteria and the associated composite suitability index (CSI).

REF#2: In short, "habitat" should be removed from the manuscript because the authors do not assess the extent or quality of habitat for any particular organism.

Authors: We cannot agree with that definition. Reviewer suggest that habitat is only limited to particular population which is more multidimensional niche space than habitat. We agree that this definition can be applied at the population level and the cited papers consider the population level definitions. However how to define the habitat for communities? Habitat according to definition should be regarded at different levels: population, species, community or biomes. Definition according to The Encyclopaedia Britannica (https://www.britannica.com/science/habitat-biology): "Habitat, place where an organism or a community of organisms lives, including all living and non-living factors or conditions of the surrounding environment. A host organism inhabited by parasites is as much a habitat as a terrestrial place such as a grove of trees or an aquatic locality such as a small pond."

For example Shearer et al . (2015) defines the habitat to given invertebrate taxa not particular organisms (*KA Shearer, JW Hayes, IG Jowett & DA Olsen (2015): Habitat suitability curves for benthic macroinvertebrates from a small New Zealand river, New Zealand Journal of Marine and Freshwater Research, DOI: 10.1080/00288330.2014.988632*).

Yujun Yiet al. (2018) (in paper: *Habitat suitability evaluation of a benthic macroinvertebrate community in a shallow lake, Ecological Indicators, https://doi.org/10.1016/j.ecolind.2018.03.039*) tried to find the relations between communities and hydrological factors. Also Theodoropoulos et al. (2018) (*Christos Theodoropoulos, Aikaterini Vourka, Nikolaos Skoulikidis, Peter Rutschmann & Anastasios Stamou (2018) Evaluating the performance of habitat models for predicting the environmental flow requirements of benthic macroinvertebrates, Journal of Ecohydraulics, 3:1, 30-44, DOI: 10.1080/24705357.2018.1440360*) tried to find the habitat relations between abundance of macroinvertebrates and BM metrics, including taxonomic richness, diversity (Shannon's index) and EPT richness (Ephemeroptera, Plecoptera, Trichoptera) and habitat preferences. The number

**of similar works can be multiplied. In google scholar phrases Habitat + suitability + macroinvertebrates are presented in 20 600 papers.**

REF#2: Similarly, the term "Weighted Usable Area (WUA)" (e.g., line 232) is meaningless without reference to the particular organisms the area is usable for. An area that is usable for one organism is not usable for some other organisms, because different species and life-history stages have different environmental requirements. And the BMWP_PL index is not a measure of the area that is usable for any organism. Therefore, the authors should also remove reference to WUA.

**Authors: According to Payne definition (PHABSIM concept)(https://www.noaa.gov/sites/default/files/legacy/document/2020/Oct/07354626138.pdf), "WUA as an index to various ecological parameters such as biomass, microhabitat area, or population size", but also macroinvertebrate abundance (Jowett 2003, Jowett IG 2003. Hydraulic constraints on habitat suitability for benthic invertebrates in gravel-bed rivers. River Research and Applications 19: 495–507. ), total macroinvertebrate biomass and ASPT biomass (Kelly et al 2015) (: DJ Kelly, JW Hayes, C Allen, D West & H Hudson (2015): Evaluating habitat suitability curves for predicting variation in macroinvertebrate biomass with weighted usable area in braided rivers in New Zealand, New Zealand Journal of Marine and Freshwater Research, DOI: 10.1080/00288330.2015.1040424)**

**The weighted usable area is defined as the total surface area having a certain combination of hydraulic conditions, multiplied by the composite probability of use for that combination of conditions. This calculation is applied to each cell within the multidimensional matrix. We calculated this area for benthic invertebrates according to BMWP_PL variation. BMWP is not a measure of area, but the index can indicate (according to its variation) which part of the habitat is optimal for invertebrates and combining those preferences with hydraulic condition we were able to estimate total surface area in both incised and redeposited rivers.**

REF#2: What the authors have done is not to assess "habitat suitability" or "weighted usable area" but rather to model the discharge that maximises the value of the BMWP_PL index. However, they have not articulated a rationale for doing so. They state that the purpose of environmental flow is to "maintain biological diversity in the river ecosystem" (lines 46-47).

**Authors: This definition is not our but we included the reference of Arthington et al. 2006.**

REF#2: But the concept of biodiversity is multi-scaled and multi-faceted - please see Rolls et al. (2018), referenced below, for an overview of the scales and facets of biodiversity relevant to environmental flows. Biodiversity is not equivalent to the value of a biotic index just because the index incorporates species with varied environmental preferences (lines 132-133).

If maintaining biodiversity is the goal. the authors need to justify their index choice by explaining which scale(s) and facet(s) of biodiversity the BMWP_PL index predicts, on what evidence, and why BMWP_PL is a better indicator of these scales and facets of biodiversity than alternative indices or metrics.

**Authors: We have never said that biodiversity is an equivalent of a biotic index. Many biotic indices inform us about the biodiversity loss e.g. BMWP, ASPT, LIFE, family biotic index etc. But they are not measures of biodiversity. We have calculated several indices, measuring taxonomic diversity (e.g. taxonomic richness, Shannon-Wiener or alpha-Fisher), but also indices related to**

**environmental perturbations (ASPT, % Oligochetae, Chironomidae/Oligochatae etc.), but BMWP index was the most sensitive to any hydrological changes.**

REF#2: I also note that while the authors have provided responses to my previous comments, in several cases they have not made corresponding changes to the manuscript. For example, the tautological phrase "multispecies communities" is still present (line 38),

**Authors: We have deleted it from the text, however in our opinion it is not tautological phrase. In Scopus database 112 articles have in their topic multispecies communities to emphasize that they are build of many species. In contrast we may have three-two species fish communities which are not multispecies at all. The concept is that multispecies communities react differently to any kind of disturbances than species-poor communities.**

REF#2: the definitions of "low low flow" and "mean low flow" (line 175) have not been incorporated,

**Authors: added to the text, we thought however that these definitions are commonly known.**

REF#2: the procedure for random selection of the case-study site is still not stated (line 179).

**Authors: added to the text**

REF#2: The authors should make changes in response to every previous comment, or else provide a reason for not making a change.

**Authors: We have provided information to every reviewer comment (some in the text or as a individual response to Reviewer) .**